# Quality assessment of integrated water vapour measurements at St. Petersburg site, Russia: FTIR vs. MW and GPS techniques

Yana A. Virolainen[1], Yury M. Timofeyev[1], Vladimir S. Kostsov[1], Dmitry V. Ionov[1], Vladislav V. Kalinnikov[2], Maria V. Makarova[1], Anatoly V. Poberovsky[1], Nikita A. Zaitsev[1], Hamud H. Imhasin[1], Alexander V. Polyakov[1], Matthias Schneider[3], Frank Hase[3], Sabine Barthlott[3], Thomas Blumenstock[3]

[1] Atmospheric Physics Department, St. Petersburg State University, Saint Petersburg, 199034, Russia
[2] Kazan (Volga Region) Federal University, 420008, Kazan, Russia
[3] Institute of Meteorology and Climate Research (IMK-ASF), Karlsruhe Institute of Technology, Karlsruhe, Germany

*Correspondence to*: Y.A. Virolainen (yana.virolainen@spbu.ru) and M. Schneider (matthias.schneider@kit.edu)

**Abstract.** The cross-comparison of different techniques for atmospheric integrated water vapour (IWV) measurements is the essential part of their quality assessment protocol. We inter-compare the synchronised data sets of IWV values measured by Fourier-transform infrared spectrometer Bruker 125 HR (FTIR), microwave radiometer RPG-HATPRO (MW) and global navigation satellite system receiver Novatel ProPak-V3 (GPS) at St. Petersburg site between August 2014 and October 2016. As the result of accurate spatial and temporal matching of different IWV measurements, all three techniques agree well with each other except for small IWV values. We show that GPS and MW data quality depends on the atmospheric conditions; in dry atmosphere (IWV smaller than 6 mm), these techniques are less reliable at St. Petersburg site than the FTIR method. We evaluate the upper bound of statistical measurement errors for clear-sky conditions as $0.29 \pm 0.02$ mm ($1.6 \pm 0.3$ %), $0.55 \pm 0.02$ mm ($4.7 \pm 0.4$ %), and $0.76 \pm 0.04$ mm ($6.3 \pm 0.8$ %) for FTIR, GPS and MW methods, respectively. We propose to use FTIR as a reference method under clear-sky conditions since it is reliable on all scales of IWV variability.

## 1 Introduction

Water vapour is one of the most important greenhouse gases in the Earth's atmosphere. Its contribution to the natural greenhouse effect is rather variable, accounting for about 70 % in clear-sky conditions (Kiehl and Trenberth, 1997). Water vapour plays a key role in different chemical processes in the atmosphere, in tropospheric dynamics, in transferring the latent heat and additionally, it is an essential part of the global hydrological cycle (IPCC, 2007). In recent years, a number of studies reported the presence of long-term trends of water vapour content in different atmospheric layers (Oltmans et al., 2000; Trenberth et al., 2005; Nilsson et al., 2008; Mieruch et al., 2008; Hegglin et al., 2014). The observed climate warming brings forward an increase of integrated water vapour (IWV), which in turn forces further climate change (Dai et al., 2012).

The importance of water vapour stimulates monitoring its content at all temporal and spatial scales with various in situ and remote sensing methods. The requirements defined for the accuracy of IWV retrieval depend on particular meteorological or climatological application area (http://www.wmo-sat.info/oscar/requirements). Thus, for example, for numerical weather

prediction and very short-range forecasts, requirements for uncertainty of measured IWV values vary from 1 mm (goal) to 5 mm (threshold). Threshold is the minimum requirement to ensure that the data are useful, goal is the "ideal" requirement above. Sometimes, these requirements are defined in relative units: statistical errors of IWV retrieval ought to be of 0.5–2.0 %, total errors – of 1–3 %. None of the existing systems for IWV measurements meets all goal requirements in terms of

uncertainty, horizontal resolution, observing cycle, timeliness. This leads to the necessity of using various methods and instruments and, consequently, fosters different programs of inter-comparison of IWV measurements with the aim to calibrate them and assess their accuracy. The latter is a part of protocol for assimilation of different measurements and numerical models data. The significant spatial (at 1–2 km scale) and temporal (at 1 hour scale) variations of IWV aggravate the problem of inter-comparison of various methods as well as their different spatial resolution, which is often evaluated only at a qualitative level.

Recently, a number of programs devoted to the inter-comparison of different methods for IWV retrieval have been performed in various geographical regions (see, for example, Palm et al., 2010; Schneider et al., 2010a; Vogelmann et al., 2011; Buehler et al., 2012; Weaver et al., 2017). The special ESA project (http://globvapour.info) was aimed at generating validated long-term satellite IWV data sets with error assessments. A book edited by Kaempfer (2013) collects results of various campaigns for ground-based water vapour retrieval. Navas-Guzman et al. (2014), Perez-Ramirez et al. (2014), and Reagan et al. (1995)

presented the typical examples of IWV inter-comparison as well. Several studies have been done in Russia recently: an inter-comparison of IWV retrieval techniques from airborne, satellite and ground-based measurements at Tomsk (Makarova et al. 2014) and series of IWV comparison by different methods at Peterhof (Semenov et al., 2015; Berezin et al., 2016, 2017; Virolainen et al., 2016; Ionov et al., 2017).

In this study, we focus on Fourier Transform Infra-Red (FTIR) technique for IWV retrieval and on the data from Bruker

125 HR spectrometer – the only instrument of such type in Russia that is certified by the Infra-Red Working Group (IRWG) of the Network for the Detection of Atmospheric Composition Change (NDACC). We analyse different types of FTIR retrievals, calculate their error budget, and propose a simple method for harmonisation of IWV values measured in different spectral regions with the aim to extend the continuous dataset of FTIR measurements. Afterward, we compare FTIR data with independent simultaneous measurements of microwave (MW) radiometer and global positioning system (GPS) receiver at St.

Petersburg site, analyse the obtained differences under various atmospheric conditions, and finally evaluate the empirical errors of considered techniques for IWV retrievals, giving the conclusions and recommendations for their usage.

In Sect. 2, we introduce St. Petersburg site for atmospheric monitoring, describe different instruments and techniques for IWV retrieval, present in detail the FTIR IWV measurements, and assess and overview the error budget of considered instrumentation and techniques. In Sect. 3, we analyse two different types of FTIR IWV retrieval, inter-compare triples of

simultaneous IWV measurements by FTIR, GPS, and MW techniques, and empirically assess the precision of studied methods. In Sect. 4, we briefly review previous studies on cross-comparison of various IWV methods, Sect. 5 summarises the most important results and conclusions of the current research.

## 2 Observational facilities at St. Petersburg site

St. Petersburg site of Saint Petersburg State University (SPbU) is located in a suburb of St. Petersburg (Peterhof) approximately 35 km west-southwest from the city centre at 59°53´ N, 29°50´ E, 20 m a.s.l. (indicated in Fig. 1 by red circle together with other European NDACC IRWG sites).

The climate of Saint Petersburg is Humid Continental (Dfb–Koeppen climate classification) with moderately mild winters and moderately warm summers. Weather is highly variable with frequent air-mass changes: approximately 165 overcast days and 140 days of cyclone activity per year. Timofeyev et al. (2016) presents a short description of the site, its instrumentation and some results of FTIR retrievals. In 2016, St. Petersburg site joined the IRWG-NDACC community with its FTIR system. In recent years, the IWV content has been regularly measured at St. Petersburg site (Semenov et al., 2015; Berezin et al., 2016,

2017; Virolainen et al., 2016; Ionov et al., 2017). Table 1 presents a description and main results of different campaigns for IWV inter-comparisons at Peterhof and gives the references to original studies. Note that all series correspond to different rather short time intervals as well as to various numerical estimates; this makes their comparison difficult. Therefore, it is worth inter-comparing all available simultaneous IWV measurements at St. Petersburg site throughout the longest available period to assess the quality of individual methods, which is the subject of the current research.

Following Semenov et al. (2014) and Berezin et al. (2016, 2017), who demonstrated that 50 km distance between locations of IWV measurements might be responsible for significant disagreement due to the spatial inhomogeneity of the water vapour fields, we exclude from the current study the radiosonde data of nearby WMO site #26063 (Voejkovo). Moreover, in the comparison we do not include the sun photometer (Cimel) measurements since they require an additional calibration procedure (Berezin et al., 2017). Therefore, the current research is devoted to the simultaneous IWV measurements by three ground-

based methods that use Bruker 125 HR spectrometer (FTIR), RPG-HATPRO radiometer (MW), and global navigation satellite system receiver Novatel ProPak-V3 (GPS).

### 2.1 FTIR method

Since the beginning of 2009, St. Petersburg site FTIR system, which consists of a Bruker 125 HR spectrometer and a self-designed solar tracker (Poberovsky, 2010), has been recording solar spectra. Atmospheric FTIR measurements using the Sun

as a light source are performed under cloudless conditions or when breaks in cloud cover allow measurements of solar spectra. The alignment of FTIR instrument is controlled by HBr cell spectra generated using both an internal light source and the Sun (Hase, 2012; Makarova et al., 2016).

We analyse the measured spectra with the PROFFIT software (Hase et al., 2004), which consists of a precise line-by-line radiative transfer model and an adaptable inversion algorithm. The latter supports different retrieval approaches (e.g. Optimal

Estimation, Tikhonov-Philips, use of logarithmised mixing ratios in the state vector, inter-species constraints for work on isotopologues). This software is routinely used at a number of NDACC sites (Kiruna, Sweden; Izana, Spain; Altzomoni, Mexico). In this study, we retrieve IWV from measurements in two different spectral regions: 1098–1222 cm$^{-1}$ (type A) and

2610–3020 cm$^{-1}$ (type M). Type A is a standard PROFFIT retrieval scheme, type M refers to the MUSICA 2015 retrieval scheme (Barthlott et al., 2017) and has a special focus on {H$_2$$^{16}$O, HD$^{16}$O/H$_2$$^{16}$O} data pairs.

Figure 2 presents typical ground-based IFS Bruker 125HR measurements of solar absorption spectra in the fitted spectral microwindows containing water vapour lines. FTIR measurements are performed with a spectral resolution of about 0.005 cm$^{-1}$

(with optical path difference of 180 cm). The A-type spectral region is characterised by the saturated water vapour lines (with the H$_2$$^{16}$O isotopologue having the stronger signatures than the HD$^{16}$O isotopologue) and their interference with O$_3$ absorption lines, whereas water vapour lines in the M-type region are of very similar line strengths for H$_2$$^{16}$O and HD$^{16}$O and not saturated but well isolated from other absorption lines. The spectral scheme for M-type retrieval also includes three micro-windows with CO$_2$ lines (in 2610-2627 cm$^{-1}$ spectral range), which are used for temperature retrieval.

To improve the IWV measurement accuracy we use approaches proposed in Schneider et al. (2010b): (a) a logarithmic scale inversion, (b) a speed-dependent Voigt line-shape model, (c) the consideration of atmospheric emission for the retrievals, (d) a simultaneous retrieval of interfering species, and finally, for the M-type retrievals, (e) a simultaneous temperature retrieval as well as the use of water vapour isotopologues inter-species constraints (Barthlott et al., 2017). The H$_2$$^{16}$O and HD$^{16}$O retrievals in the A-type setup are made independently. For spectroscopic parameters of the absorption lines, we use the

HITRAN2008 database with 2009 updates (Rothman et al., 2009) with slight modifications of pressure broadening and line intensities: for the spectral range used by the A-type retrieval, according to Schneider et al. (2011), and for spectral range used by the M-type retrieval, according to the Appendix of Barthlott et al. (2017).

The corresponding pressure and temperature profiles used for the analysis are the daily National Center for Environmental Prediction (NCEP) re-analysis data (Lait et al., 2005) for Peterhof location. The a priori profiles of interfering atmospheric

constituents are adopted from The Whole Atmosphere Community Climate Model (WACCM) data – using a single set of climatological a priori profiles during all seasons for Peterhof location (Park et al. 2013). The selected mode of the PROFFIT retrieval code is based on the Tikhonov–Phillips approach (A-type) (Tikhonov 1963; Phillips 1962) and on the optimal estimation method (M-type) (Rodgers 2000), respectively. After analysis of spectra, we filtered the IWV retrievals, depending on the ratio: remaining measurement noise (in percent) to the number of dofs (degrees of freedom for signal). Further, we

consider only those retrievals, for which this ratio is less than 1.0 for the A-type and 0.5 for the M-type retrievals. We chose this criterion in accordance with the signal-to-noise ratio in corresponding spectral regions, trying to optimise the number of less noisy measurements that represent all atmospheric conditions. Thus, for the period between March 2009 and December 2016 we selected 3265 and 3548 IWV retrievals of A-type and M-type, respectively.

For the error budget calculations, we assume the same uncertainty sources and values for both types of the retrieval. We

calculate "gain matrices" that show the sensitivity of the retrieval to some error source, and associated covariance matrices for statistical and systematic errors (Rodgers 2000). For calculating the error matrices, the uncertainty of the error sources are taken into account as listed in Table 2. The relative weight of statistical and systematic contribution to the total IWV error varies depending on the error source (Hase et al., 2004). The spectroscopic line parameters uncertainty (see Table 2) is the major source of systematic errors. In the A-type spectral channels (see Fig. 2) the intensity of solar radiation is smaller than in

the M-type, resulting in a decrease of signal-to-noise ratio and, consequently, in an increase of measurement noise and its influence on statistical errors of the A-type retrieval. Statistical errors of the A-type retrieval are controlled mainly by temperature profile uncertainty, whereas in the M-type scheme temperature profile is simultaneously retrieved together with the target gases. Thus, our estimations indicate that IWV retrieval of the M-type is slightly more precise than that of the A-type due to: a) the higher signal-to-noise ratio, and b) the simultaneous temperature retrieval.

## 2.2 MW method

The 14-channel microwave radiometer RPG-HATPRO (generation 3) is one of the instruments used for IWV measurements at St. Petersburg site. It has been functioning since June 2012 in constant mode with sampling interval of about 1–2 s and the integration time of 1 s. A complete description of radiometers of the RPG-HATPRO type is presented at the website of the manufacturer (http://www.radiometer-physics.de). All information relevant to the experimental setup can be found in the paper by Kostsov et al. (2016). It should be noted that IWV measurements are performed from zenith observations only.

We analyse the atmospheric MW radiation brightness temperature spectra by two separate and independent retrieval algorithms. The first algorithm is the built-in regression algorithm (REGR) provided by the manufacturer and tuned for SPbU measurement site. The algorithm uses a quadratic regression scheme applied to the brightness temperature observations in zenith mode plus surface pressure sensor data. Tuning of this algorithm is based on radiative transfer calculations for atmospheric models that have been compiled using 10 years of radio-sounding data at Voejkovo station near St. Petersburg. The absolute accuracy of IWV retrievals by the REGR algorithm declared by the manufacturer is of 0.3 mm, the random noise is less than 0.05 mm. Our estimates of IWV variations in stable atmospheric conditions for different days of measurements (Virolainen et al., 2016) showed that the standard deviation (SD) of means for RPG HATPRO totals 0.05–0.09 mm, which is rather close to noise level presented by other researchers (Steinke et al., 2015).

The second algorithm is based on the inversion of the radiative transfer equation and therefore is referred below as "physical algorithm" (PHYS). This algorithm uses the well-known and widely applied approach of simultaneous retrieval of several atmospheric parameters profiles, which influence the radiative transfer at frequencies corresponding to spectral channels of the MW radiometer. Since the problem is ill-posed, we used the optimal estimation method for its regularisation. The description of the specific features of the physical algorithm applied to RPG-HATPRO measurements, assessment of the retrieval accuracy for different parameters, and the examples of retrievals can be found in the paper by Kostsov (2015a). Besides the brightness temperature measurements, the PHYS algorithm utilizes the surface pressure, temperature and humidity readings, the temperature and relative humidity profile statistics as well as the hydrostatic equilibrium constraint by applying the general approach to the solution of multi-parameter inverse problems (Kostsov, 2015b). To calculate the IWV values, we integrate the absolute humidity vertical profile. We obtain the IWV retrieval error from the error matrix corresponding to the absolute humidity profile, which we calculate for every single set of brightness temperature measurements. Therefore, the IWV retrieval error estimate is a variable quantity. In practice, the values of statistical retrieval error estimates for the PHYS algorithm are within the interval 0.08–0.10 mm.

The MW IWV retrievals at St. Petersburg site considered in earlier studies (Berezin et al., 2016, 2017; Virolainen et al., 2016; Ionov et al., 2017) correspond to the REGR algorithm. In this study, we use the PHYS algorithm to analyse the MW measured spectra due to the following reasons: a) the PHYS algorithm provides the error estimates for every single retrieval together with a quality control flag, which is very useful for detection and removal of spurious data; b) the output of PHYS algorithm

5 is a self-consistent set of several atmospheric parameters (water vapour and temperature profiles, pressure profile, and cloud liquid water content); and c) the PHYS algorithm is a flexible tool that gives the possibility to use different amount of input data and it is more convenient for modifications, if necessary.

We compared the results of the REGR and PHYS retrieval algorithms through the whole period of measurements (2012–2016) to assess the differences in retrieved IWV values. Relative mean difference of the two data sets does not depend on absolute

10 IWV values: REGR is biased high compared to PHYS by approximately 5 %, which means that there is a factor of 1.05 between two retrieval techniques. Almost a half of the absolute differences (PHYS vs. REGR) are between -0.6 and -0.2 mm. The mean difference between two data sets amounts to -0.52 mm with the SD of 0.44 mm. The two retrievals are less consistent in a dry atmosphere. The RPG-HATPRO radiometer is operating at its limits below 5 mm of IWV, which is affected by the intrinsic relative weakness of the 22 GHz water vapour line. Therefore, the errors of both methods are increasing with

15 decreasing IWV values in dry conditions. Differences in the IWV results of the two algorithms might be due to many reasons, particularly to different a priori information, radiative transfer model, etc.

### 2.3 GPS method

GPS method implies a technique of active remote sensing by GPS satellites, which transmit radio signals in the microwave range. Before these signals can reach the Earth-based receiver, they are delayed and refracted in the atmosphere. Owing to the

20 permanent dipole moment of water vapour molecule, atmospheric refractivity is very sensitive to the presence of this gaseous constituent in the atmosphere (Businger et al., 1996). Since a hydrogen bond between water molecules in liquid water and ice significantly reduces the contribution of the dipole moment to radio signal delay, the impact of cloud water and ice on atmospheric refractivity is limited. This allows ground-based GPS receivers to provide data on IWV above the receiver site even in cloudy weather. If the position of the receiver is accurately known, a target atmospheric delay is derived by comparison

25 between an observed signal path length (pseudorange) and a geometric distance between satellite and receiver (true range).

For IWV retrieval, we use a ground-based GPS sensor – Novatel ProPak-V3 dual-frequency receiver with a GPS-702-GG antenna mounted on a roof. The instrument is operating continuously in all weather conditions since August 2014. The carrier phase and binary code pseudorange measurements from GPS satellites on two GPS carrier frequencies (1227 and 1575 MHz) are processed with the help of TropoGNSS software, developed at Kazan Federal University (Kalinnikov and Khutorova,

30 2017). Retrieval algorithm identifies the Precise Point Positioning strategy for zenith tropospheric delay (ZTD) estimation (Kouba, 2009). Phase measurements play a key role in the algorithm: they are compared with geometric distances between receiver and corresponding satellites. Code measurements serve only for calculation of receiver clock corrections. Geometric distances and satellite clock corrections are determined using precise ephemerides/clock products of International GNSS

Services (http://www.igs.org). The algorithm takes into account changes of a receiver antenna position due to the ocean loading effect, solid and pole tides (Petit and Luzum, 2010). Influence of ionosphere is excluded by the formation of iono-free combination of phase measurements at two frequencies (Schaer, 1999). Phase ambiguities are removed by forming differences between phase measurements from two consecutive epochs. Slant tropospheric delays during measurements are expressed in

the form of multiplication of zenith tropospheric delay and Niell mapping function that are determined by the zenith angle of each satellite, day of a year, latitude and altitude of station (Niell, 1996). The zenith cut-off angle in TropoGNSS processing is established at 83°. Time series of iono-free combinations of phase measurements are consistently processed from epoch to epoch by Kalman filter with ZTD as unknown parameter. Output ZTD time series have a 5-minute step. We assume that ZTD is a sum of the dry hydrostatic (ZHD) and the wet (ZWD) components (Bevis et al., 1994). The hydrostatic component with

the accuracy of 1 mm is determined using Saastamoinen model (Saastamoinen, 1972). The wet component is defined as a difference between ZTD and ZHD and then is converted to IWV values following an approach proposed by Askne and Nordius (1987) and Mendes (1999).

The uncertainty of ZHD and ZWD determination results in the IWV retrieval uncertainty of 1.5–2 mm. This estimate is close to the uncertainty of GPS IWV measurements, obtained by other authors. Ning et al. (2016) reported the uncertainty of IWV

measurements of around 0.5–1.0 mm; Steinke et al. (2015) presented the uncertainty of GPS IWV measurements of about 1–2 mm. Ionov et al. (2017) assessed the errors of various methods of IWV retrieval at St. Petersburg site by analysing the differences in simultaneous IWV retrievals and got the following estimates: the statistical error of FTIR and MW measurements made up 0.3 mm, while for GPS technique it constituted 0.5 mm.

The variability of presented errors of IWV measurements can be explained by the dependence of retrieval errors on the

atmospheric state, particularly, the IWV values, measurement conditions (solar zenith angles, number of used satellites, etc.), the stability of instruments and the consistency of measurements themselves. Taking into account the location of St. Petersburg site between the Gulf of Finland and rural areas, local horizontal gradient of water vapour distribution might be also a reason for differences.

## 3 IWV measurements at St. Petersburg site

All instruments for IWV monitoring are installed at the buildings of SPbU Peterhof campus: RPG-HATPRO radiometer and Novatel ProPak-V3 receiver – on a roof of the same building (at a distance of 2 m), 55 m a.s.l.; Bruker 125 HR spectrometer – in the ground floor of a nearby building at a distance of 330 m to the west, 21 m a.s.l. Figure 3 depicts schematically a diagram of mutual location of all three instrumentation. It is worth mentioning that the solar tracking system of Bruker 125 HR is located on a roof, so the beam path partly passes through a pipe from the top of the building to the ground floor. In accordance

with the beam pattern of MW radiometer, input signal comes from about 20 m above the instrument. We evaluated possible differences in measured IWV values due to differences in an elevation of considered instrumentation. We used the ECMWF monthly averaged humidity profiles and got the following estimates. Depending on season, FTIR technique might give values

0.1–0.4 mm and 0.2–0.5 mm higher than GPS and MW techniques, respectively. The difference between GPS and MW might reach 0.1–0.2 mm.

Although MW radiometer and GPS receiver are located close to each other, they have some spatial disagreement: MW radiometer is operated only in a zenith observation mode for IWV measurements, whereas GPS receiver gets the information from various satellites with a horizontal averaging of several dozen of kilometres. We also should take into account the difference in observed air masses while comparing MW and FTIR measurements. Virolainen et al. (2016) have demonstrated that depending on season and time of the day (different solar azimuth and zenith angles), the measured IWV values may belong to different air mass located at a distance of up to 20–25 km. At worst, the spatial inhomogeneity of water vapour fields might cause the discrepancy between two types of measurements, especially, considering the surroundings of St. Petersburg site – the Gulf of Finland from one side and the rural suburbs of Saint Petersburg from the other.

MW radiometer measures spectra every 2 s, GPS receiver gets a single measurement every 5 minutes, FTIR spectrometer records spectra only in clear-sky conditions, one record usually lasts about 12 minutes. Table 3 lists the main features of considered instrumentation for IWV measurements. It is worth mentioning that the period of observations of all three instruments varies from device to device: FTIR has been working since January 2009, MW radiometer since June 2012, and GPS receiver since August 2014. There are also some gaps in measurement series due to technical problems with one or more instruments. In order to synchronise all three types of IWV measurements, we average MW and GPS measurements over 12-minutes interval for each FTIR individual measurement. In this study, we consider the period of IWV measurements from September 2014 to December 2016 when such triples are available.

### 3.1 FTIR measurements

In earlier papers (Semenov et al., 2014; Virolainen et al., 2016; Berezin et al., 2017; Ionov et al., 2017), we presented the results of FTIR IWV retrievals that used the spectral scheme close to the A-type. In this study, we add the M-type retrieval developed for the MUSICA project (Schneider et al., 2016; Barthlott et al., 2017). Therefore, we compare FTIR IWV retrievals of these two setups to harmonise our previous and present results.

Since we record solar spectra within limited spectral bands using a set of broadband filters, the spectra underlying the A-type and M-type retrievals are not observed simultaneously. The acquisition time for individual interferograms obtained by co-adding ten scans equals approximately 12 minutes. We usually make a series of three individual measurements for each spectral band. Thus, there is a time lag between the two types of FTIR IWV measurements of at least 12–15 minutes.

In order to compare data sets of the A-type and M-type IWV measurements we assume the pairs to be near-synchronised if the time mismatch between the nearest ones does not exceed 30 minutes. The number of near-synchronised pairs totals 820 for the whole period of considered FTIR IWV measurements. Figure 4 depicts the IWV time series measured by FTIR spectrometer at St. Petersburg site in the period of 2009–2016. Relative differences between the M-type and A-type retrievals as well as mean differences in absolute and relative units and the correlation coefficient are shown in the bottom of Fig. 4. A dominant factor, which influences systematic differences between the A-type and M-type retrievals and does not have a

pronounced seasonal or intra-annual dependence in relative units, is presumably a difference in spectroscopic line parameters and their accuracy in the two spectral regions (Rothman et al., 2009).

Figure 5 (left) illustrates the correlations between the M-type and A-types retrievals. The slope demonstrates an overestimation of the M-type retrieval vs. A-type retrieval with a factor of $1.09\pm0.02$. Buehler et al. (2012) obtained the slope of 1.06,

comparing FTIR measurements in nearly the same spectral regions for Kiruna site (Sweden). Taking into account the uncertainties of the both slopes, we may conclude that they are very close. Consequently, the assumption of 2% uncertainty for water vapour line intensities might be too optimistic and hence systematic errors might be larger than those indicated in Table 2. Actually, the adjustment of the $H_2^{16}O$ and $HD^{16}O$ M-type line parameters is based on a few aircraft profile measurements. Schneider et al. (2016) estimated the confidence of the $H_2^{16}O$ and $HD^{16}O$ line parameter adjustments to be

about 10%, meaning that the used line intensities might have an uncertainty of up to 10% (please, be aware that the adjustment aimed on $\{H_2^{16}O, HD^{16}O/H_2^{16}O\}$ pairs and for the $H_2^{16}O/HD^{16}O$ ratios the confidence is better than 1.5%).

Figure 5 (right) depicts a histogram showing a relative frequency of absolute differences between the M-type and A-type retrievals: the distribution of differences is close to a lognormal probability distribution (likewise the distribution of IWV values themselves) with location parameter of -0.06 and scale parameter of 0.80. Almost a half of differences are between 0.25

and 1.25 mm.

Table 4 lists statistical characteristics of the M and A-type comparison, depending on IWV values. Absolute differences increase with growing IWV values, relative differences slightly fall in the 8.6-9.9 % range. To harmonise IWV measurements of the A-type and M-type, we multiply IWV values of the A-type by a factor of 1.09 and add an offset of 0.14 mm (and get so-called the $A_{corr}$-type retrieval). The observed mean difference between the M-type and $A_{corr}$-type and the standard deviation

reduce to zero and to 0.42 mm (3 %), respectively. The standard deviation value is within the error margins of both types of retrievals (see Table 2), so we may conclude that both setups agree well. Therefore, for the following analysis and comparison with independent IWV measurements we combine data sets of the M-type and $A_{corr}$-type to cover a more extended period.

Figure 6 shows an example of using the $A_{corr}$-type retrieval in harmonisation and analysis of IWV diurnal cycle by the FTIR-method. The number of separate measurements of one type may be insufficient to detect the strong variation of IWV in contrast

with using a combination of both types of FTIR data. Figure 6 (right) demonstrates that considering only a M-type retrieval, we miss decreasing IWV values up to 9–10 mm (8–9 am 13.09.2016). At the same time, the correction of an A-type measurement helps to avoid "artificial" IWV variations caused by systematic differences between the retrieval types (Fig. 6, left).

## 3.2 Simultaneous FTIR, MW and GPS measurements

Finally, we create three data sets of synchronised IWV measurements: FTIR (M-type + $A_{corr}$-type), MW (PHYS) and GPS for the period between August 2014 and October 2016. Figure 7 depicts results of comparison as scatter plots showing correlation between the data pairs. Generally, different data sets correlate well, the correlation coefficient is close to or larger than 0.99 for considered data pairs. However, the scatter of IWV values obtained from different techniques depends on IWV values

themselves, the smaller the values the greater the scatter. We observe the smallest IWV values obtained by GPS technique (less than 1 mm). The accuracy of GPS measurements makes up 0.5–1.5 mm and worse (see Sect. 2.3), thus for dry atmosphere, the errors of GPS technique might be larger than measured IWV values (more than 100%). At the same time, measurement errors of MW technique are also larger for small IWV values due to the weakness of the 22 GHz water vapour line (see Sect.

2.2). The best agreement between data pairs is observed for IWV values larger than 5–6 mm. Even for these IWV values, FTIR measurements agree better with GPS and MW data than GPS and MW with each other.

Table 5 presents results of the same comparison (mean differences and their standard deviations) of all IWV data pairs, i.e. shows biases and scatters between the different techniques. Since GPS and MW techniques are less accurate for small IWV values, we single out two subsets depending on IWV quantity: less than 6 mm ("dry" subset) and larger than 6 mm ("wet"

subset). FTIR and GPS measurements are in better agreement than other considered data pairs (the smallest scatter, the strongest correlation), whereas GPS and MW experience the largest scatter of differences with minimal bias. For all pairs, the smallest scatter in absolute and relative units is observed for the subset with IWV values greater than 6 mm. A percentage scatter for the "dry" subset varies from 14.4 to 27.1 %, whereas for the "wet" subset it ranges from 4.6 to 7.1 %. The worst agreement belongs to GPS–MW pairs. These values of scatter and correlation coefficient confirm that in dry atmosphere GPS

and MW techniques are less reliable for IWV measurements at St. Petersburg site than the FTIR method.

Taking FTIR measurements as a reference, for the whole dataset and for the "wet" subset, we observe an underestimation of GPS and MW data, with larger dry bias for the latter. The same situation is for "dry" FTIR–GPS pairs. Particularly, this systematic discrepancy can be explained by differences in instruments elevations above sea level (last column of Table 5) discussed at the beginning of Sect. 3. However, it is not the only reason for a systematic disagreement of IWV values, since

the observed differences are larger than estimated for different elevation (Table 5). On the contrary, for the "dry" subset, the bias between FTIR–MW and GPS–MW techniques is quite different: GPS measurements have a dry bias and FTIR measurements have no bias compare to MW data. This probably results from increasing errors of MW measurements in a dry atmosphere. Taking into account differences in elevation of considered instrumentation and corresponding estimates in IWV differences, we may reduce a wet bias of FTIR measurement with respect to GPS data up to 0.2 mm, not depending on observed

IWV values. In this context, biases in pairs with MW measurements depend on IWV values themselves. Thus, for the whole data set and the "wet" data set, we may reduce a dry bias of MW measurements up to 0.2–0.3 and 0.4–0.6 mm in comparison with GPS and FTIR measurements, respectively. For small IWV values (<6 mm), the dry MW bias converts into a wet bias estimated as 0.2 and 0.4 mm compare to FTIR and GPS data.

### 3.3 Empirical statistical assessment of IWV measurement errors

Having three co-located methods for IWV retrieval at our disposal, we may empirically evaluate the uncertainty of individual methods. The individual estimate of IWV value measured by method $A$ $x_i^A$ can be expressed as $x_i^A = x_{i,true}^A + M^A + \sigma_i^A$, where $x_{i,true}^A$ is the true value of IWV, $M^A$ and $\sigma_i^A$ are the systematic and statistical errors, respectively. Taking into account

spatial $\sigma^{space}$ and temporal $\sigma^{time}$ misalignments of the two types of measurements (*A* and *B*) and assuming that statistical measurement errors are uncorrelated and have a zero mean, we can express the square of the observed standard deviation $\sigma^{A-B}$ as follows:

$$\sigma^{A-B\,2} = \overline{\sigma^{A\,2}} + \overline{\sigma^{B\,2}} + \overline{\sigma^{time\,2}} + \overline{\sigma^{space\,2}} \tag{1}$$

Since we inter-compare three near-synchronised data sets, we may assume that the temporal misalignment is equal to zero. As we have noted earlier, FTIR spectrometer tracks the Sun, MW radiometer has a zenith-viewing geometry, and GPS-receiver gets the information from different satellites providing a spatial-averaged value of IWV. Therefore, the considered data triples might have a spatial disagreement (Virolainen et al, 2016). However, we do not have two-dimensional maps of IWV fields at our disposal, thus we cannot make a quantitative estimation of spatial disagreement. At the same time, we cannot select a

statistically significant dataset of dates with small IWV variability to analyse measurement uncertainties without spatial disagreement – we observe only 8 days in considered period with 1% IWV variability during daytime. Therefore, we neglect this misalignment error, too. It means that we evaluate the upper bound of statistical measurement errors. Using Eq. (1) for each pair of data sets, we obtain a system of three linear equations, from which we can derive the empirical statistical errors for each of the compared methods:

$$\sigma_{FTIR} = \sqrt{\frac{1}{2}(\sigma^2_{FTIR-MW} + \sigma^2_{FTIR-GPS} - \sigma^2_{GPS-MW})}$$

$$\sigma_{MW} = \sqrt{\frac{1}{2}(\sigma^2_{FTIR-MW} + \sigma^2_{GPS-MW} - \sigma^2_{FTIR-GPS})}$$

$$\sigma_{GPS} = \sqrt{\frac{1}{2}(\sigma^2_{GPS-MW} + \sigma^2_{FTIR-GPS} - \sigma^2_{FTIR-MW})} \tag{2}$$

Using standard deviation values from Table 5 in these equations, we get the statistical errors for the whole data set of compared IWV measurements (see the second column of Table 6). Since differences between considered measurements strongly depend

on IWV values (see Sect. 3.2), we tried to get the same estimates for the "dry" and "wet" subsets. The third column of Table 6 displays errors for the "wet" subset, while for the "dry" subset the system of Eq. (2) could not be solved presumably due to correlations of measurement errors of one or more instruments (MW, GPS) in dry atmosphere. Comparing the whole (819 triples) and "wet" (658 triples) data sets results, we see that errors significantly decrease only for MW technique; along with the largest among other instruments error values, it confirms that MW measurements are less reliable in dry atmosphere.

For verification of the quality and accuracy of our estimates, we allocate one more data set: – the so-called "MW stable" data set, for which we select measurements with variability of MW IWV values less than 2% for 12 minutes averaging interval (717 data triples). The estimates for "MW stable" data set are shown in the fourth column of Table 6. All estimates from Table 6 allow us to assess the uncertainty of empirical statistical errors of IWV measurements at St. Petersburg site as $0.29 \pm 0.02$ $(1.6 \pm 0.3)$, $0.55 \pm 0.02$ $(4.7 \pm 0.4)$, and $0.76 \pm 0.04$ mm $(6.3 \pm 0.8\ \%)$ for FTIR, GPS and MW methods, respectively. Ionov

et al. (2017) reported for FTIR and MW methods empirical statistical errors of 0.3 mm, for GPS of 0.5 mm. The estimates for FTIR and GPS techniques are nearly the same as in the current study, whereas for MW method the error doubles. The difference

between the two studies probably lies in different time samples and different types of interpretation of MW and FTIR spectra measurements.

Summing up the results of the IWV retrieval accuracy assessment, we may conclude that at St. Petersburg site FTIR and GPS techniques demonstrate more stable and consistent results than MW technique.

## 4 Discussion

A great number of studies are devoted to the analysis of differences in water vapour FTIR measurements caused by differences in retrieval schemes: spectral microwindows, algorithms, a priori information, etc. (Schneider et al., 2010a, 2010b; Schneider and Hase, 2009; Sussmann et al., 2009; Palm et al., 2010). Schneider et al. (2010b) compared the vertical profiles and total columns of $H_2O$ and HDO in 790–880, 1090–1330, 2650–3180, and 4560–4710 cm$^{-1}$ spectral regions and showed that minimal statistical IWV measurement errors had been observed for shortwave spectral interval 4560–4710 cm$^{-1}$ due to the largest signal-to-noise ratio. The correction of spectral line parameters (line intensities and half widths) allowed reducing the systematic errors up to 1 % under assumption of 1 % uncertainty in line intensities (Schneider and Hase, 2009, Schneider et al., 2011). FTIR measurements underestimated radiosonde data systematically with differences of 0.1–0.5 % for 2650–3180 and 1090–1330 cm$^{-1}$ and 2.2–2.4 % for 4560–4710 and 790–880 cm$^{-1}$ spectral regions (Schneider et al., 2010b). The standard deviation of means amounted to approximately 7 % in all cases. It is worth mentioning that IWV values measured in different spectral regions are in good agreement. Schneider et al. (2010b) estimated the differences between FTIR IWV retrievals in 1090–1330 and 2650–3180 cm$^{-1}$ spectral regions to be of -1.0 % with the standard deviation of 1.2 %; Buehler et al. (2012) reported differences of -3.4 ± 7.3 %. We observe larger differences between the A-type and M-type schemes; the mean reaches -8.9% (see Table 4). At St. Petersburg site, IWV ranges from below 1 mm to more than 40 mm, and both the mean difference and its standard deviation are very consistent for different IWV values indicating the high quality of IWV variability as obtained from both retrieval schemes. To extend the data set of FTIR measurements we correct the A-type retrievals in accordance with the M-type IWV values and use this joint data set for comparisons with independent measurements.

The spatial mismatch of compared data sets might significantly influence the results of the comparison. Semenov et al (2015) coupled radiosonde data in Voejkovo with FTIR data at Peterhof; Berezin et al (2016) did the same for MW data. Although correlations between measurements are higher than 0.96, root mean square differences reached more than 20% for most of the collocated IWV data sets. The strong disagreement was mainly due to the natural spatial variability of IWV, taking into account the distance of 50 km between Peterhof (MW and FTIR instruments location) and Voejkovo (radiosondes launches). This variability reached approximately 13 mm during a day (Semenov et al., 2015). Even for monthly means of FTIR and radiosonde data (correlations higher than 0.99), the SD values reached approximately 11 % (or 0.98 mm). Excluding the days with strong IWV variability allowed reducing mean differences between Voejkovo and Peterhof measurements up to 3–4%, the SD values up to 12–14 %. Vogelmann et al. (2015) analysed spatial and temporal variations of water vapour by FTIR and lidar measurements and indicated their major role in discrepancies of different methods. They observed the strong spatial (at

1–4 km distance) and temporal (at 5–15 minutes interval) variations up to 0.35 mm in summer. Steinke et al. (2015) also indicated spatial inhomogeneity of IWV at 8–10 km scale as the reason of 0.6 mm statistical differences between different methods. Even nearly the same location of two instruments (MW and FTIR), but different viewing angle may influence the results of inter-comparison, if observed air masses are at a distance of up to 20–25 km (Virolainen et al., 2016).

Table 7 presents examples of statistical results for FTIR, MW and GPS methods inter-comparisons at a number of ground-based measurement stations. It is worth noting that direct comparisons of FTIR and MW IWV methods are very few. MW radiometers used in inter-comparison campaigns (Buehler et al., 2012; Palm et al., 2010) were originally designed for observations of other atmospheric species; IWV was derived as a by-product, so the IWV retrieval scheme as well as its accuracy had not been optimised. Buehler et al. (2012) reported differences between MW and FTIR data sets at Kiruna site

(Sweden), which varied from (-1.90 ± 12.85) % up to (22.79 ± 29.34) %, or from (-0.20 ± 0.92) mm up to (0.90 ± 1.08) mm. Virolainen et al. (2016) showed that at St. Petersburg site (Russia) MW measurements overestimated FTIR data by 0.29 mm, not depending on season. The SD values varied from 0.24 mm (the dry season) to 0.54 mm (the wet season), amounting to 0.42 mm (4.1 %) in average. In this study, differences between the two types of measurements are larger and have opposite sign, amounting to (0.85 ± 0.87) mm for FTIR vs. MW measurements. Such different results for the same site can be explained

by different FTIR retrieval schemes: the A-type in earlier study (Virolainen et al., 2016) and the M-type in this study. We have shown that both types of retrieval have systematic difference in IWV values of about 9 %, which can be easily corrected by a proposed simple harmonisation scheme.

   The comparison between FTIR and GPS IWV data sets are discussed in several studies (Schneider et al., 2010a; Buehler et al., 2012; Mengistu et al., 2015). For both Arctic and African site, GPS measurements overestimate FTIR data by 0.3–0.6 mm.

The standard deviation of mean differences varies from 0.9 to 1.6 mm. Our estimates at St. Petersburg site demonstrate nearly the same in percent underestimation of GPS vs. FTIR data as at Izana site (Canary islands), but larger in absolute values. This wet bias of FTIR measurements at St. Petersburg site may come from the location of GPS sensor being 34 m higher than FTIR spectrometer, which might be crucial under specific atmospheric conditions.

   Finally, many studies are devoted to comparisons of GPS and MW IWV measurements (van Baelen et al., 2005; Memmo et

al., 2005; Morland et al., 2006; Buehler et al., 2012; Steinke et al., 2015; Roman et al., 2016). Mean differences between GPS and MW methods vary from -2.63 to 1.36 mm with standard deviations up to 3.69 mm. Buehler et al. (2012) explained such significant differences by the influence of clouds and, probably, precipitation on the accuracy of MW method. Ning et al. (2016) reported a detailed error analysis of GPS method, indicating total errors of GPS method to be equal to 0.6–0.7 mm for 23–33 mm of IWV values. Roman et al (2016) presented a number of GPS and MW comparisons for IWV together with an

inter-comparison of independent GPS measurements at different measurement sites in the framework of the Atmospheric Radiation Measurement (ARM) program. In general, for all stations, GPS data overestimate MW measurements. For ARM NSA (North Slope of Alaska) site, which is similar to St. Petersburg site with respect to IWV variability range, systematic and random differences are much smaller than for ARM TWP (Tropical Western Pacific) and SGP (Southern Great Plains) stations. It is worth mentioning that for TWP site, the disagreement between the two GPS data sets reached (0.8 ± 3.1) mm. The

differences between GPS and MW data at St. Petersburg site are very similar to those reported by Roman et al. (2016) for ARM NSA site.

Most of the differences presented in Table 7 (between FTIR, MW and GPS IWV data pairs) are larger than observed in this study at St. Petersburg site. The stringent spatial and temporal matching conditions applied here is the predominant reason, in

our opinion, of good agreement of different methods for IWV measurements. Our figures demonstrate that with an accurate spatial and temporal matching of different types of IWV measurements their disagreements are close to the total measurement errors of individual methods and thereafter to WMO goal requirements to the accuracy of IWV measurements in atmospheric chemistry.

## 5 Summary

The variety of requirements for IWV measurements in different fields of atmospheric science, its strong spatial and temporal variability leads to the fact that there is no single unique method for IWV measurements that meets all requirements for accuracy, periodicity of measurements, horizontal resolution, etc. We describe three methods for IWV measurements (FTIR, MW, and GPS) available at St. Petersburg site and compare these observations as near-synchronised data triples.

We focus on the FTIR technique checking whether it can be used as a reference for MW and GPS methods under clear-sky

conditions at the new NDACC site. We analyse the MUSICA IWV retrievals (M-type) in comparison with the standard PROFFIT retrieval (A-type) to enable the comparison with results of previous studies. We evaluate averaged IWV measurement errors for the whole period of measurements (2009–2016) from the error matrix calculations and demonstrate that the M-type retrieval is slightly more accurate (systematic errors constitute 2.0 vs. 2.3 %) and precise (statistical errors make up 0.4 vs. 0.9 %) than the A-type retrieval. We observe the overestimation of the M-type retrieval vs. A-type retrieval

with a scaling factor of $1.09 \pm 0.2$. The mean difference between the M-type and A-type retrievals amounts to $(1.2 \pm 0.8)$ mm or $(8.9 \pm 5.9)$ % and is mainly caused by the different spectroscopy in spectral regions related to the A-type and M-type setups. We harmonise M-type and A-type of IWV retrievals to increase the continuity of a series of IWV measurements by FTIR method at St. Petersburg site. The correction of A-type retrievals by a factor of 1.09 and by adding offset of 0.14 mm allows reducing the differences between the M-type and A-type data to $(0.0 \pm 0.4)$ mm or $(0 \pm 3)$ %, which is close to the IWV

measurement errors of the FTIR method. We may recommend to use such harmonisation at other sites equipped with high-resolution FTIR spectrometers.

We analyse in detail FTIR, MW and GPS techniques for IWV retrieval at St. Petersburg site and allocate the data triples of near-synchronised IWV measurements by all three methods. We show that FTIR and GPS measurements are in better agreement among all coincident pairs, whereas GPS and MW methods experience largest scatter of differences with minimal

bias. FTIR vs. GPS methods agree within $(0.59 \pm 0.62)$ mm or $(5.0 \pm 5.3)$ %, FTIR vs. MW – within $(0.91 \pm 0.86)$ mm or $(7.8 \pm 7.3)$ %, and finally GPS vs. MW – within $(0.33 \pm 0.97)$ mm or $(2.9 \pm 8.8)$ % for the whole data set of synchronised

triples. It is worth mentioning that in a dry atmosphere (IWV values less than 6 mm) FTIR method is more reliable for IWV measurements than MW or GPS techniques, for which the measurement errors are increasing with decreasing IWV values.

We observe an underestimation of GPS and MW techniques with respect to FTIR data that occurs particularly due to differences in elevation of considered instruments (GPS sensor is located higher than Bruker 125HR by 34 m, MW – by 54 m).

Accounting for differences in IWV values due to the different elevation of instruments may significantly reduce systematic discrepancies between FTIR, GPS and MW IWV measurements at St. Petersburg site. Horizontal inhomogeneity of water vapour fields in the vicinity of the observing site might also result in the discrepancy of compared quantities due to different observational geometry, since FTIR spectrometer tracks the Sun, while MW radiometer has a zenith-viewing geometry, and GPS-receiver gets the information from different satellites providing a spatial-averaged value of IWV.

Moreover, we empirically evaluate the upper bound of statistical measurement errors for all three methods and get the following estimates for clear-sky conditions at St. Petersburg site: $0.29 \pm 0.02$ ($1.6 \pm 0.3$), $0.55 \pm 0.02$ ($4.7 \pm 0.4$), and $0.76 \pm 0.04$ mm ($6.3 \pm 0.8$ %) for FTIR, GPS and MW techniques, respectively. We demonstrate that MW method is less consistent in IWV retrieval, especially under dry atmospheric conditions, presumably due to an operational instability of the MW radiometer RPG-HATPRO at St. Petersburg site. Nevertheless, all three techniques agree well with each other and

therefore are suitable for monitoring of IWV values at St. Petersburg site.

We compare our estimates with the published results and assume that accurate spatial and temporal matching of different IWV data is necessary for achieving a good agreement between measurements within the measurement errors of individual methods and thereafter with WMO goal requirements to the accuracy of IWV measurements.

We conclude that FTIR method is highly accurate, but applicable only under clear-sky conditions; MW and GPS are all-

weather methods, but are less reliable in dry atmosphere. Therefore, we cannot recommend any instrument or technique as the best choice for the networks measuring IWV under variety of atmospheric conditions. The different observation techniques complement each other rather than outperform each other. Based on our results, we propose to use FTIR as a reference method under clear-sky conditions since it is reliable on all scales of IWV variability.

*Acknowledgement.* The experimental part of the study has been supported by the Russian Foundation for Basic Research (grants number 15-05-07524 and 16-05-00681). The processing and analysis of the data have been performed with the financial support of the Russian Science Foundation (grant number 14-17-00096). The MUSICA FTIR retrievals have been developed in the framework of the MUSICA project, which has been funded by the European Research Council under the European Community's Seventh Framework Programme (FP7/2007-2013) / ERC Grant agreement number 256961.

The Centre for Geo-Environmental Research and Modelling "GEOMODEL" of Saint-Petersburg University provides the measurement facilities. J.W. Hannigan (NCAR, Boulder, CO, USA) kindly provides the station-specific WACCM data. We also acknowledge the availability of the NCEP data. The GPS-measurements have been interpreted in accordance with the Russian Government Program of Competitive Growth of Kazan Federal University.

We acknowledge support by Deutsche Forschungsgemeinschaft and Open Access Publishing Fund of the Karlsruhe Institute of Technology.

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

**Table 1. The IWV comparison campaigns at St. Petersburg site, involving the data of FTIR, GPS, sun photometer Cimel, and radiosonde (R/s) measurements. Mean is the mean difference, SD is the standard deviation of mean difference.**

| Pairs | Period | Mean $\pm$ SD, mm (%) | Correlation coefficient | Reference |
|---|---|---|---|---|
| FTIR–R/s | April 2009–March 2012 | -0.06 $\pm$ 2.34 (-1.7 $\pm$ 22.1) | 0.961 $\pm$ 0.007 | Semenov et al., 2014 |
| R/s–MW | March 2013–May 2014 | -0.31 $\pm$ 3.13 (-2.2 $\pm$ 22.1) | 0.934 $\pm$ 0.004 | Berezin et al., 2016 |
| MW–FTIR | March 2013–June 2015 | 0.29 $\pm$ 0.42 (2.8 $\pm$ 4.1) | 0.9982 $\pm$ 0.0001 | Virolainen et al., 2016 |
| MW–Cimel | March 2013–May 2014 | 1.56 $\pm$ 1.07 (10.9 $\pm$ 7.4) | 0.9962 $\pm$ 0.0001 | Berezin et al., 2017 |
| FTIR–Cimel |  | 1.11 $\pm$ 0.94 (10.1 $\pm$ 8.6) | 0.9952 $\pm$ 0.0003 |  |
| R/s–Cimel |  | -0.5 $\pm$ 2.6 (-4 $\pm$ 19) | 0.944 $\pm$ 0.005 |  |
| MW–GPS | September 2014–June 2015 | -0.3 $\pm$ 0.6 (-3 $\pm$ 9) | 0.997 $\pm$ 0.001 | Ionov et al., 2017 |
| FTIR–MW |  | -0.4 $\pm$ 0.4 (-7 $\pm$ 8) | 0.995 $\pm$ 0.002 |  |
| FTIR–GPS |  | -0.5 $\pm$ 0.6 (-6 $\pm$ 11) | 0.998 $\pm$ 0.002 |  |

**Table 2. Error budget of IWV retrieval by the FTIR technique.**

| Error source | Uncertainty | A-type errors, % | | M-type errors, % | |
|---|---|---|---|---|---|
| | | Statistical | Systematic | Statistical | Systematic |
| Baseline (offset and channelling) | 0.1 % and 0.2 % | 0.3 | 0.3 | 0.3 | 0.3 |
| ILS (modulation efficiency and phase error) | 1 % and 0.01 rad | 0.08 | 0.08 | 0.06 | 0.06 |
| Temperature profile | 1° K (surface–10 km a.s.l.) 1° K (10–35 km a.s.l.) 1° K (above 35 km a.s.l.) | 0.8 | 0.3 | 0.04 | 0.02 |
| Spectroscopy ($S$ and $\gamma$) | 2 % and 2 % | – | 2.2 | – | 2.0 |
| Measurement noise | From residuals | 0.2 | – | 0.1 | – |
| Total error, % | | $0.9 \pm 0.3$ | $2.3 \pm 0.3$ | $0.4 \pm 0.2$ | $2.0 \pm 0.2$ |

**Table 3. Instrumentation for IWV measurements at St. Petersburg site.**

| Acronym | Instrumentation | Geometry | Comments |
|---------|-----------------|----------|----------|
| FTIR | Bruker IFS-125HR | Solar-tracking | Clear-sky, 12 min – duration |
| GPS | Novatel ProPak-V3 | 20-30 km horizontal resolution | Day and night, every 5 min |
| MW | RPG-HATPRO | Zenith-viewing | Day and night, every 2 s |

**Table 4. Statistics of synchronised IWV pairs (M-type–A-type) for different IWV values. Relative differences (mean ± SD) in percent have been calculated with respect to the M-type values. R is the correlation coefficient.**

| IWV limits, mm | Matches | Difference | | R |
|---|---|---|---|---|
| | | mm | % | |
| <5 | 165 | 0.3 ± 0.1 | 9.9 ± 4.2 | 0.99 ± 0.01 |
| 5–10 | 164 | 0.7 ±0.3 | 9.6 ± 4.2 | 0.98 ± 0.02 |
| 10–15 | 146 | 1.2 ± 0.5 | 9.2 ± 3.6 | 0.95 ± 0.03 |
| 15–20 | 172 | 1.5 ± 0.4 | 8.8 ± 2.4 | 0.95 ± 0.02 |
| >20 | 173 | 2.2 ± 0.6 | 8.6 ± 2.3 | 0.99 ± 0.01 |
| all | 820 | 1.2 ± 0.8 | 8.9 ± 5.9 | 0.999 ± 0.002 |

**Table 5. Statistics of differences (mean ± SD) for all coincident IWV pairs depending on IWV values. Relative values correspond to the first instrument in pairs. R is the correlation coefficient. Elevation bias is an estimated difference due to the difference in an elevation of the instruments. Number of measurements totals 819, 161, and 658 for 0–45 mm, 0–6 mm, and 6–45 mm data sets, respectively.**

| Pairs | IWV, mm | Difference | | R | Elevation bias, |
|---|---|---|---|---|---|
| | | mm | % | | mm |
| | 0–45 | 0.59 ± 0.62 | 5.0 ± 5.3 | 0.995 ± 0.003 | |
| FTIR–GPS | <6 | 0.37 ± 0.54 | 9.9 ± 14.4 | 0.94 ± 0.03 | 0.1–0.4 |
| | >6 | 0.64 ± 0.63 | 4.7 ± 4.6 | 0.993 ± 0.005 | |
| | 0–45 | 0.91 ± 0.86 | 7.8 ± 7.3 | 0.994 ± 0.004 | |
| FTIR–MW | <6 | 0.00 ± 0.59 | 1.3 ± 15.7 | 0.92 ± 0.03 | 0.2–0.5 |
| | >6 | 1.13 ± 0.77 | 8.2 ± 5.7 | 0.991 ± 0.005 | |
| | 0–45 | 0.33 ± 0.97 | 2.9 ± 8.8 | 0.988 ± 0.005 | |
| GPS–MW | <6 | -0.32 ± 0.92 | -9.6 ± 27.1 | 0.81 ± 0.05 | 0.1–0.2 |
| | >6 | 0.49 ± 0.92 | 3.7 ± 7.1 | 0.983 ± 0.007 | |

**Table 6. Empirical statistical errors of FTIR, GPS and MW methods, as obtained from Eq. (2) for different data sets. Estimates are given in mm, in parentheses in percent.**

| Data set | 0–45 mm | 6–45 mm | MW-stable |
|----------|---------|---------|-----------|
| FTIR | 0.29 (1.6) | 0.27 (1.3) | 0.31 (1.9) |
| GPS | 0.55 (5.1) | 0.57 (4.4) | 0.52 (4.6) |
| MW | 0.80 (7.1) | 0.72 (5.5) | 0.75 (6.3) |

**Table 7.** Some results of IWV comparisons as reported by different authors. SD is the standard deviation of mean differences; R is the correlation coefficient.

| Study | Location | Matches | Bias, % (mm) | SD, % (mm) | R |
|---|---|---|---|---|---|
| FTIR vs. MW | | | | | |
| Buehler et al., 2012 | Kiruna, Sweden | 54–104 | 1.9– -22.8 (0.2– -0.9) | 12.8–29.3 (0.92–1.08) | 0.933– 0.986 |
| Virolainen et al., 2016 | Peterhof, Russia | 1254 | -2.8 (-0.29) | 4.1 (0.42) | 0.998 |
| Current study | Peterhof, Russia | 819 | 7.8 (0.91) | 7.3 (0.86) | 0.994 |
| FTIR vs. GPS | | | | | |
| Buehler et al., 2012 | Kiruna, Sweden | 1329–1473 | -7.4– -10.9 (-0.29– -0.61) | 14.8–16.0 (0.91–1.02) | 0.982– 0.983 |
| Schneider et al., 2010a | Izana, Spain | 112 | 5.36 (0.09) | 19.5 (0.73) | 0.958 |
| Mengistu et al., 2015 | Addis Ababa, Ethiopia | 113 | (-0.6) | (1.6) | 0.92 |
| Current study | Peterhof, Russia | 112 | 5.0 (0.59) | 5.3 (0.62) | 0.995 |
| MW vs. GPS | | | | | |
| Van Baelen, 2005 | Toulouse, France | 60–65 | (0.51–1.36) | (2.60–3.21) | |
| Memmo et al., 2005 | Elba, Italy | 1831 | (0.01) | (1.3) | 0.986 |
| Morland et al., 2006, | Bern, Switzerland | | (0.5) | (1.0) | |
| Buehler et al.,2012 | Kiruna, Sweden | 640–1385 | 10.2–12.9 (0.45–0.93) | 27.0–28.6 (1.75–2.66) | 0.867– 0.931 |
| Steinke et al, 2015 | Jülich, Germany | 3859 | (0.18) | (0.91) | 0.99 |
| Roman et al., 2016 | See text for details | SGP–1055 | (-1.04) | (1.81) | 0.990 |
| | | TWP–181 | (-2.63) | (3.69) | 0.961 |
| | | NSA–8116 | (-0.41) | (0.79) | 0.999 |
| Current study | Peterhof, Russia | 819 | -2.9 (-0.33) | 8.8 (0.97) | 0.988 |

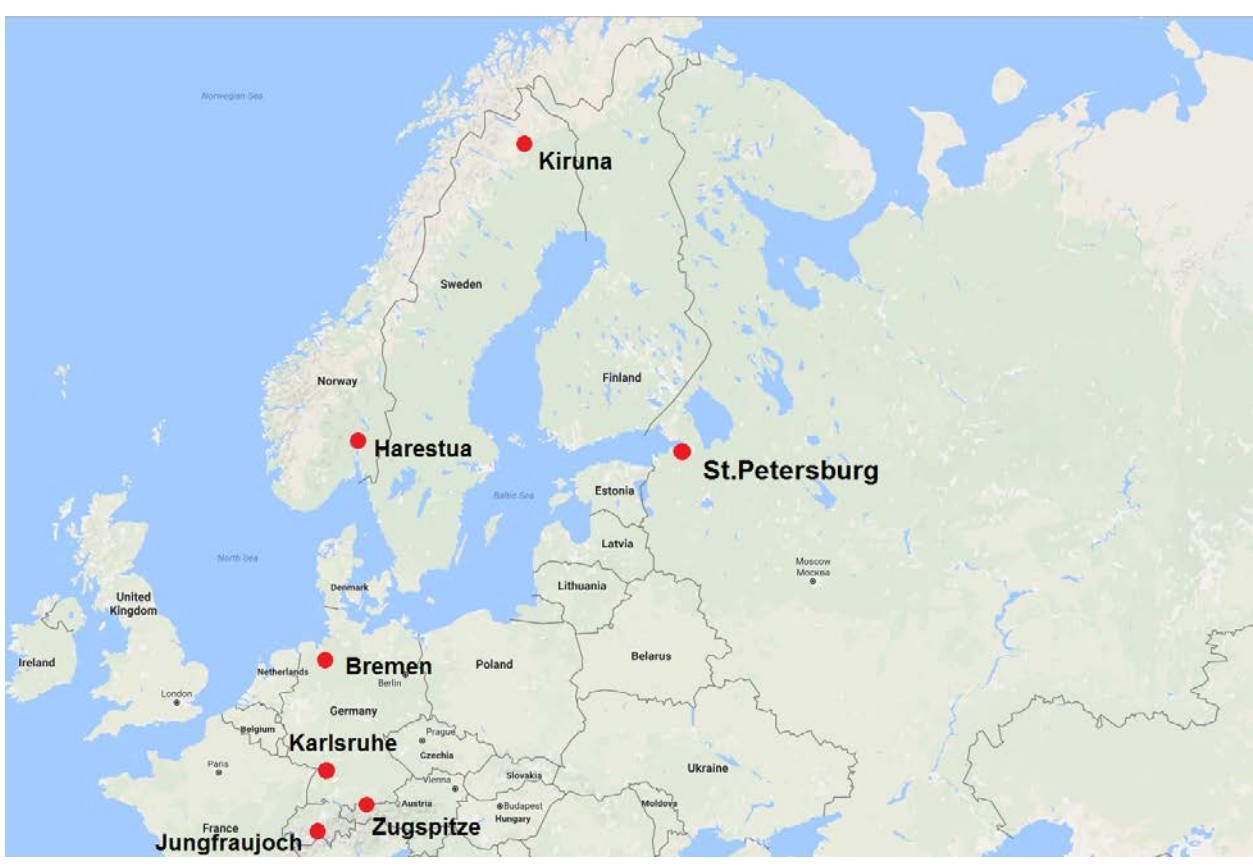

10    **Figure 1. Location of NDACC IRWG sites in Europe (red circles).**

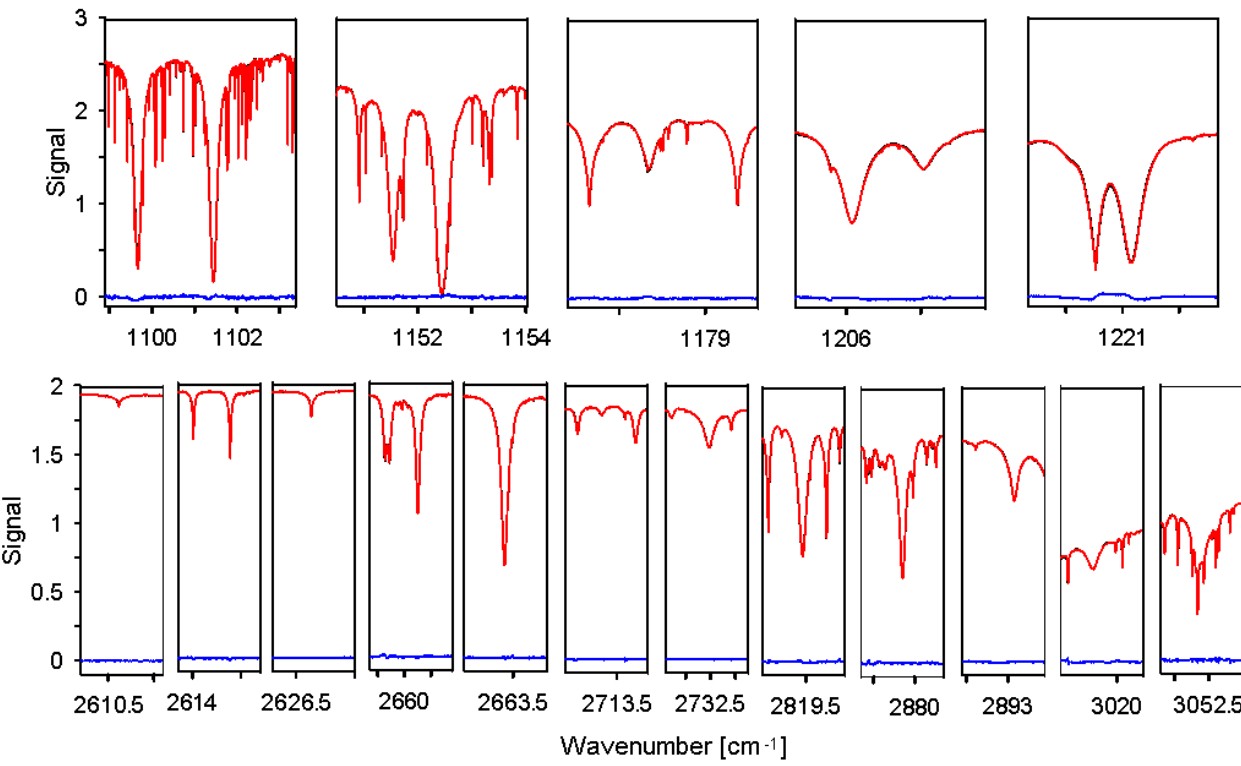

**Figure 2. The A-type (above) and M-type (below) spectral microwindows taken from spectra measured on the 7ᵗʰ of May 2016. Red lines: measured spectra, black lines: simulated spectra, blue lines: residuals (difference between measurement and simulation). IWV content equals 8 mm, solar elevation angle – 47.1°.**

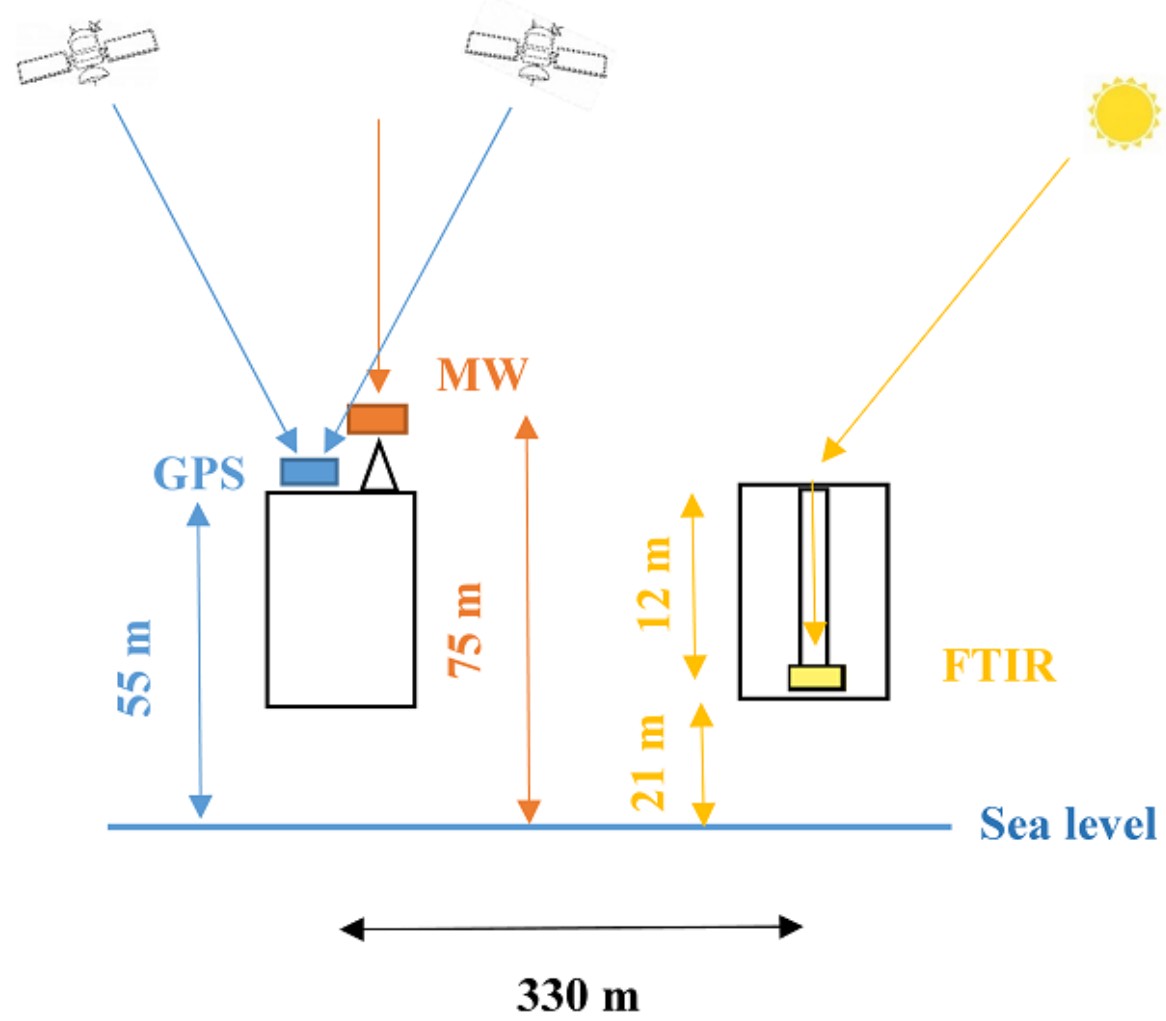

Figure 3. Schematic location of FTIR, MW, and GPS instrumentation at St. Petersburg site.

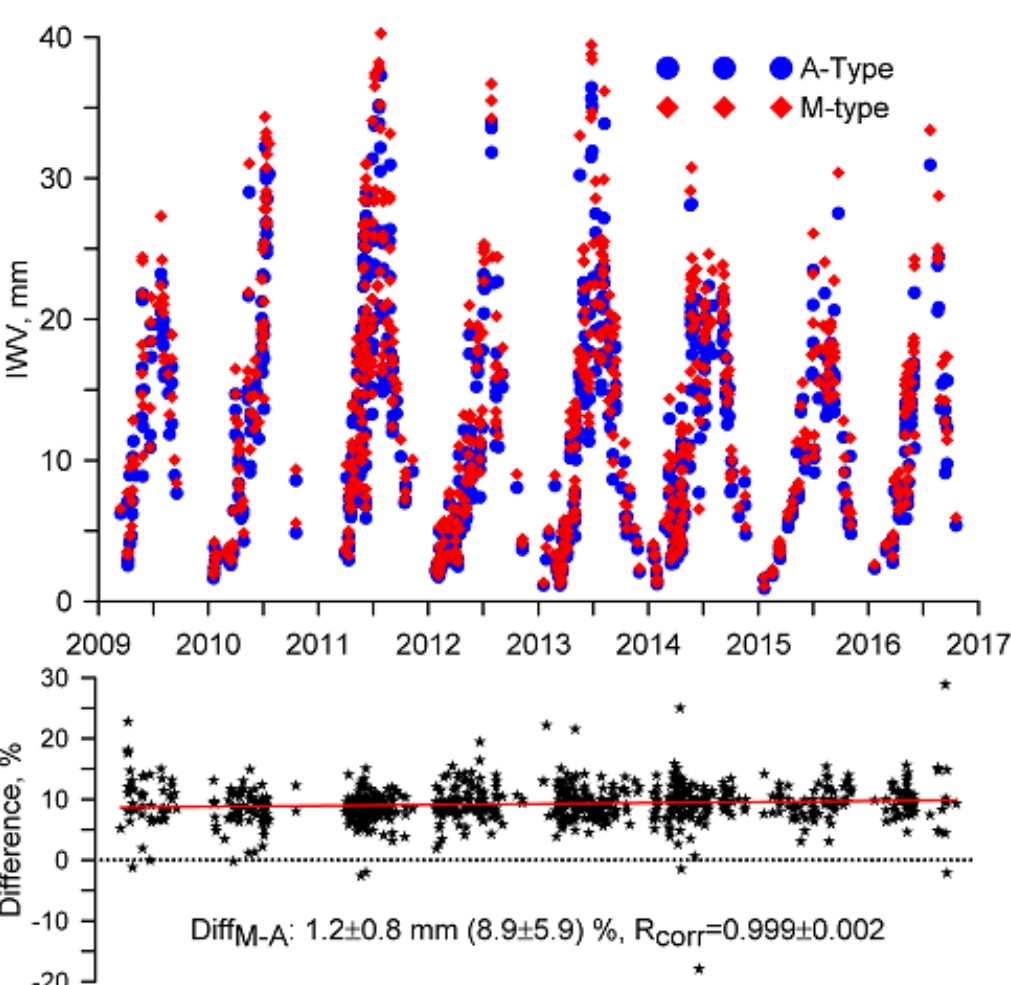

**Figure 4. Time series of IWV measurements (above) by FTIR method: the A-type (blue circles) and M-type (red diamonds) and their relative difference (below). Red line indicates a time dependence of differences approximated by a linear regression line.**

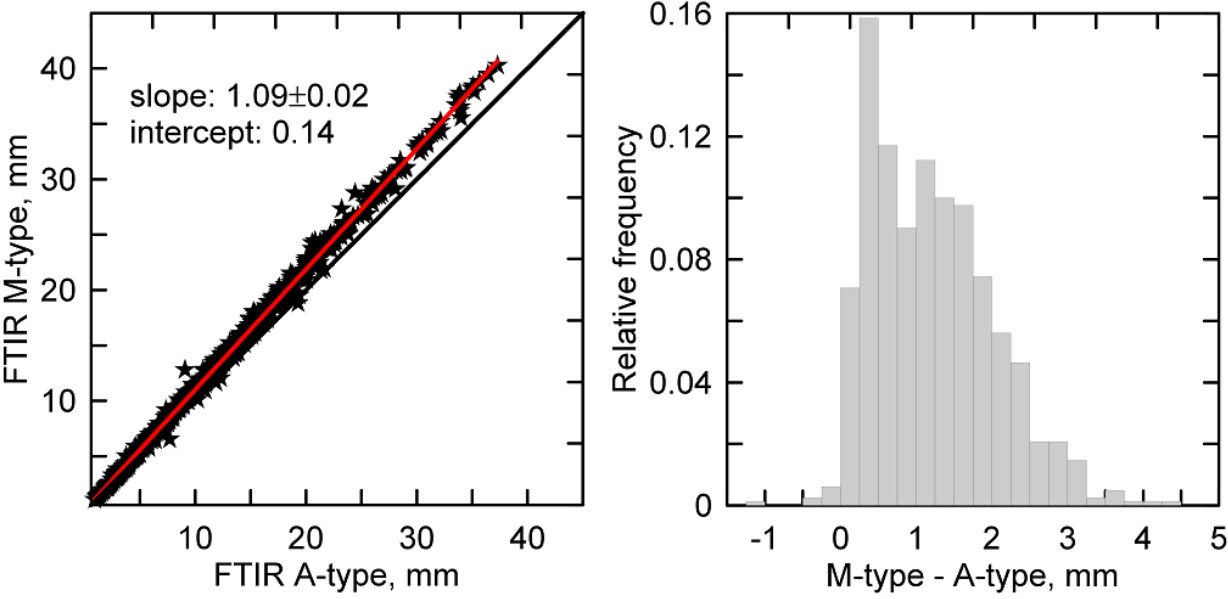

**Figure 5. Correlations of synchronised IWV pairs: the M-type vs. A-type (left), histogram of distribution of absolute differences: the**
10   **M-type minus A-type (right). Red line is the linear regression line.**

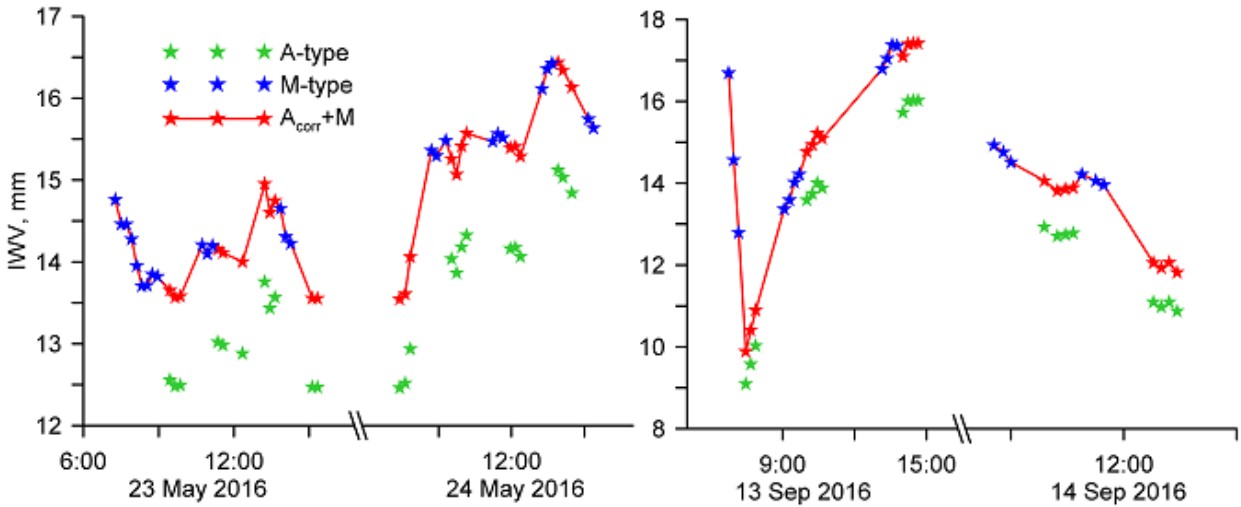

**Figure 6. Diurnal variations of IWV content observed by FTIR technique: the A-type retrieval (green stars), the M-type retrieval (blue stars), and the joint A$_{corr}$ and M-type retrieval (red line with stars).**

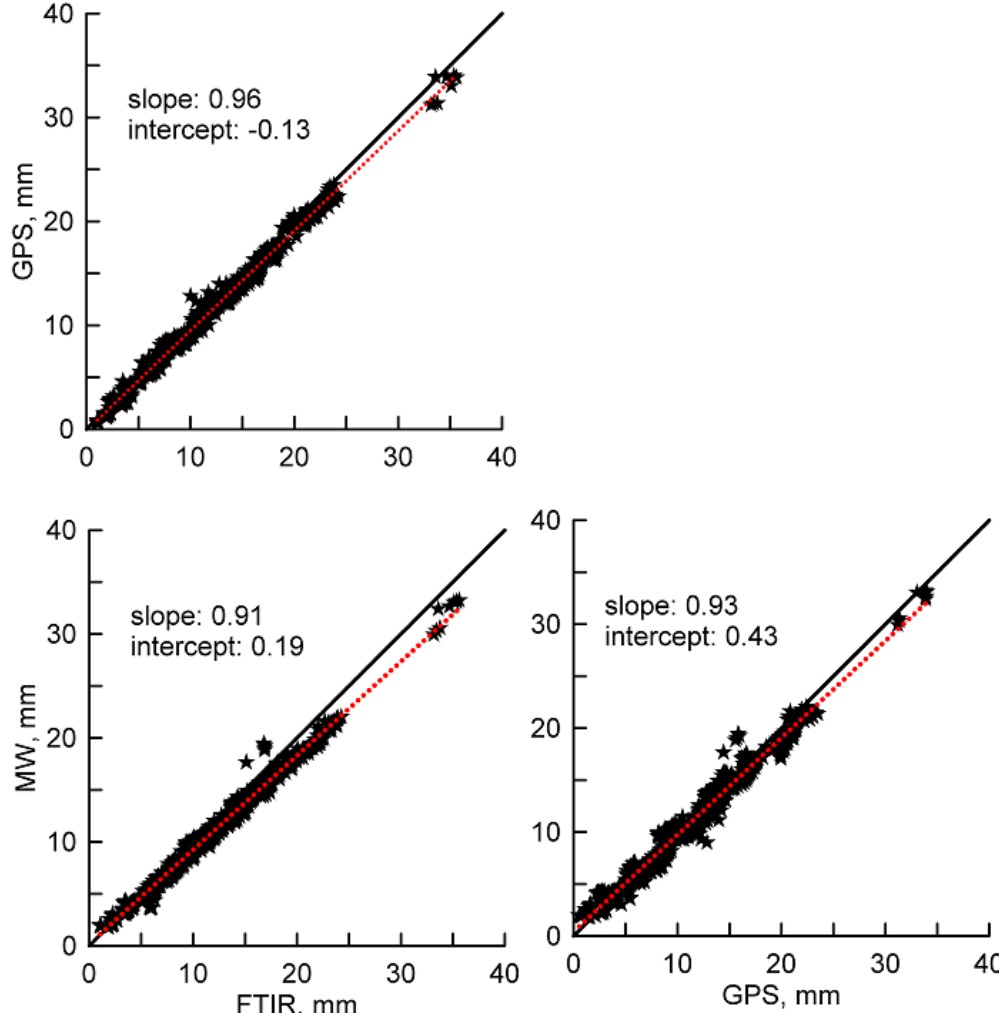

**Figure 7. Correlation of IWV values measured by FTIR, GPS and MW techniques. Red dotted line is the linear regression line. The slope and intercept of linear regression line are given in each panel.**