# Peer review of "Quality assessment of integrated water vapour measurements at St. Petersburg site, Russia: FTIR vs. MW and GPS techniques"

_Atmospheric Measurement Techniques, 2017_

## Referee Comment (RC1) · Anonymous Referee #1 · 28 Jun 2017

The manuscript describes an assessment of 3 remote sensing techniques for measuring integrated water vapor. These measurements span different but overlapping time periods and are located at the Peterhof NDACC site. They look at specific instrument comparisons and statistical comparisons for the ensemble. They investigate wet/dry biases, effect of distance and time of measurements. They explore two methods for the FTIR retrieval and find a robust correlation that can be used to remove the small bias. The conclusions show excellent agreement among the instruments consistent with similar investigations.

Overall this is an excellent paper very nearly ready for publication. This reviewer found

only one point that should have some clarification. Pg 4 line 24 the use of the ratio measurement noise to DOFS (also DOFS should be capitalized) with a cutoff of unity as a criterion for acceptable retrievals is not universal or necessarily intuitive and requires some definition or rationalization.

This manuscript represents excellent and complete work.

———————————————

---

## Referee Comment (RC2) · Anonymous Referee #2 · 7 Sep 2017

The paper is well written and organized with a very good background discussion that helps put the work into historical perspective with past efforts of a similar nature. The work itself is well done and follows similar past work. This is the weakness of the paper because the work follows so closely to Virolainen et al., 2016 only with a different locale that allowed inclusion of the GPS data. No new methods for the comparisons are proposed in the current work nor additional insight into the problem is noted. Thus, the paper is a modest extension of past work. I recommend the work be published with the hope that the authors will consider including a few statements clearly highlighting how the current work goes beyond the current literature.

[Figure]

The comments below are suggestions I think would help the paper, but I leave it to the authors to decide whether they need to be implemented.

The abstract does a good job of summarizing the work including quantitative results.

The paper's background is very thorough and the requirements for IWV retrieval accuracies and limitations of the various methods help give the paper perspective.

The paper does an excellent job of referencing past work that helps put the current work into perspective. In fact, too good of a job since the current paper follows very closely to Virolainen et al. 2016.

Methods are well described and follow past techniques, thus no technical issues in the current work.

In figure 5, I disagree that the comparison plot needs to be in log scale in order to cover IWV from 1-30 mm. The log scale will always make the fit look appreciably better than it is. At the very least, there should be some indication of the goodness of fit to straight line. This appears to some extent in the tables and follow-on discussions but it would be better to include this information with the figure. The 0.02 standard deviation of the fit and the large number of data indicate that the slope is statistically different from the a 1:1 line, but a direct comment to this effect would be helpful. Also included in the statistical discussion should be whether the slope and standard deviation of the current method are truly different from the 1.06 slope retrieved by Buehler. At first glance, there does not seem to be a statistical difference.

An interpretation of the log-normal result for the A to M retrieval difference should be given as well as the logic for using only a multiplicative correction given that the offset of the fit is 0.14 mm. It does not seem as though using the full fit would increase the complexity of an A to M harmonization.

Again, I disagree with the use of a log-log scale for Figure 7. More important is that the labeling on Figure 7 does not make it clear which IWVs are being compared. The

text allows me to infer that the lower right is MW versus GPS but the other two are not easily figured out from the text. It would be better to label both the x and y axes.

From the results presented, it is not clear that that the FTIR is the clear better answer for low IWVs. It is clear that the GPS disagrees with the MW and FTIR data, but it could also be concluded that the MW data performs better at low IWV since the variability of both the FTIR-MW and FTIR-GPS data show similar values. This could imply that the FTIR is the root of cause of the large variability. The authors should provide clearer justification for why they conclude FTIR is the method of choice for low IWV.

The approach used in Section 3.3 is not providing an accuracy assessment as much as a relative uncertainty between the three methods. The terminology used in the summary as it being an empirically based upperbound of statistical measurement errors is much more correct.

For the uncertainty assessment, selection of dates for which the IWV varied by small amounts (<1% for example) would imply a spatial homogeneity and would provide data for which the assumption of spatial homogeneity would be better than simply neglecting the misalignment error. The authors should point out if there are not enough dates with little variability or why the uncertainties from those dates are not suitable for the assessment of the overall uncertainty.

The paper provides a wealth of information, both from the current work and reference to past efforts. The difficulty for the reader is locating the key important points of the new results. For instance, are the authors recommending that FTIR users correct A and M retrieval differences and do they propose their approach as the best method? Are the estimated uncertainties acceptable, do they agree with past work, are there recommendations for which method is more suitable? These questions are covered with the results in the paper, but are not easily found.

One last comment is that the authors have an opportunity to discuss whether all three methods can be viewed as equally suitable for IWV and do not. That is, do the authors recommend that a network of IWV instrumentation rely on a single measurment approach or can it use any of the three methods? Is the capability of MW and GPS measuring under cloudy conditions worth the added scatter in the measurments? Is a zenith-viewing MW measuremnt with higher scatter better than a solar-path based FTIR measurement with lower scatter? Including such insight into the paper would improve its relevance to the broader community and take it past being only another collection of IWV retrieval results.

---

## Author Comment (AC1) · 22 Sep 2017

Author response to reviewer's comments on "Quality assessment of integrated water vapour measurements at St. Petersburg site, Russia: FTIR vs. MW and GPS techniques" by Yana A. Virolainen et al.

Reply to Anonymous Referee #1

First of all, we would like to thank the reviewer for reading and commenting the manuscript.

"The manuscript describes an assessment of 3 remote sensing techniques for measuring integrated water vapor. These measurements span different but overlapping time periods and are located at the Peterhof NDACC site. They look at specific instrument comparisons and statistical comparisons for the ensemble. They investigate wet/dry biases, effect of distance and time of measurements. They explore two methods for the FTIR retrieval and find a robust correlation that can be used to remove the small bias. The conclusions show excellent agreement among the instruments consistent with similar investigations. Overall this is an excellent paper very nearly ready for publication. This reviewer found only one point that should have some clarification. Pg 4 line 24 the use of the ratio measurement noise to DOFS (also DOFS should be capitalized) with a cutoff of unity as a criterion for acceptable retrievals is not universal or necessarily intuitive and requires some definition or rationalization. This manuscript represents excellent and complete work."

We used such criterion after analysis of all measured spectra. The idea was to exclude "noisy" spectra that can influence the quality of water vapour measurements from the following analysis. If we take into account only remaining measurement noise, we can lose measurements with large solar zenith angle (SZA), especially during wintertime. Thus, we took a ratio to the number of dofs, which usually increases with increasing SZA. The specific value of this ratio was chosen for both retrieval schemes in accordance with the optimum between two requirements: to exclude "noisy" measurements and to save as much representative measurements in each season as possible. That is why the value of the criterion differs for different spectral schemes. In both cases, we filtered out about 10-13% of all measurements. We added a sentence to the text trying to clarify this item.

---

## Author Comment (AC2) · 22 Sep 2017

Author response to reviewer's comments on "Quality assessment of integrated water vapour measurements at St. Petersburg site, Russia: FTIR vs. MW and GPS techniques" by Yana A. Virolainen et al.

Reply to Anonymous Referee #2

First of all, we would like to thank the reviewer for helpful comments and suggestions to the manuscript.

RC: "The paper is well written and organized with a very good background discussion

that helps put the work into historical perspective with past efforts of a similar nature. The work itself is well done and follows similar past work. This is the weakness of the paper because the work follows so closely to Virolainen et al., 2016 only with a different locale that allowed inclusion of the GPS data. No new methods for the comparisons are proposed in the current work nor additional insight into the problem is noted. Thus, the paper is a modest extension of past work. I recommend the work be published with the hope that the authors will consider including a few statements clearly highlighting how the current work goes beyond the current literature."

AC: The previous paper by Virolainen et al., 2016 was mainly descriptive; there were the first attempts of comparing and analysing the differences of two types (MW and FTIR) of integrated water vapour (IWV) measurements under various atmospheric and viewing conditions at Peterhof site. In the current paper, we focus on FTIR technique checking whether it can be used as a reference for MW and GPS methods under clearsky conditions at the new NDACC site for atmospheric monitoring. The advantage of FTIR technique is the high accuracy of IWV measurements on all scales of its variability. Thus, we analyse in details all available FTIR measurements, assess theoretically their accuracy, and propose a scheme for harmonisation of IWV measurements obtained in different spectral ranges with the aim to get a more complete and continuous dataset for further comparisons. The proposed harmonisation also can be used in analysis of IWV diurnal variations as well as a priori information for water vapour as interfering gas in monitoring of other atmospheric gases. Therefore, the harmonisation of FTIR IWV measurements is one of the key points of the current research. The extension of period for IWV measurements at St. Petersburg site in comparison with Virolainen et al., 2016 allows us to add GPS method, which is widely available at many ground-based locations. Thus, our assessment of GPS IWV uncertainty can be useful to other research groups. Moreover, the increase and expansion of datasets as well as the more detailed and careful analysis in the current paper allows to get more reliable results and conclusions. One more key point of the current research is the empirically based assessment of the upper bound of statistical measurement errors of three dif-
ferent techniques for IWV measurements that allows us to make conclusions about the reliability and uncertainty of all considered methods, discussing their advantages and disadvantages under various atmospheric conditions. Some of these conclusions are common, some are site related. We tried to highlight these key points in the Introduction section.

RC: "The comments below are suggestions I think would help the paper, but I leave it to the authors to decide whether they need to be implemented. The abstract does a good job of summarizing the work including quantitative results. The paper's background is very thorough and the requirements for IWV retrieval accuracies and limitations of the various methods help give the paper perspective. The paper does an excellent job of referencing past work that helps put the current work into perspective. In fact, too good of a job since the current paper follows very closely to Virolainen et al. 2016. Methods are well described and follow past techniques, thus no technical issues in the current work."

AC: We thank the reviewer for evaluating the structure of the paper and for suggesting ideas to improve the manuscript. We tried to note the differences of the current research and paper by Virolainen et al., 2016 above and in the text of the current paper.

RC: "In figure 5, I disagree that the comparison plot needs to be in log scale in order to cover IWV from 1-30 mm. The log scale will always make the fit look appreciably better than it is. At the very least, there should be some indication of the goodness of fit to straight line. This appears to some extent in the tables and follow-on discussions but it would be better to include this information with the figure. The 0.02 standard deviation of the fit and the large number of data indicate that the slope is statistically different from the a 1:1 line, but a direct comment to this effect would be helpful. Also included in the statistical discussion should be whether the slope and standard deviation of the current method are truly different from the 1.06 slope retrieved by Buehler. At first glance, there does not seem to be a statistical difference. An interpretation of the log-normal result for the A to M retrieval difference should be given as well as the logic
for using only a multiplicative correction given that the offset of the fit is 0.14 mm. It does not seem as though using the full fit would increase the complexity of an A to M harmonization."

AC: We choose a logarithmic scale due to the large variability of IWV values (two orders of magnitude). The distribution of IWV values can be approximated by a lognormal distribution, thus a log-scale presentation adequately shows how the scatter depends on IWV values. Nevertheless, to clear doubts we changed the scales to linear and added information about the quality of linear fit to Fig.5. We used only multiplicative correction because by this correction we removed the difference in spectroscopic lines parameters - the dominant factor of that difference, which did not depend on the atmospheric and measurement conditions. The reason for the remaining offset of 0.14 mm is not so obvious. Nevertheless, we treated this offset, too and got slightly different results of comparison with other techniques.

RC: "Again, I disagree with the use of a log-log scale for Figure 7. More important is that the labeling on Figure 7 does not make it clear which IWVs are being compared. The text allows me to infer that the lower right is MW versus GPS but the other two are not easily figured out from the text. It would be better to label both the x and y axes."

AC: We changed the scales to linear for Figure 7 as well. Sorry for labelling absence – it was misprint, and now it is fixed.

RC: "From the results presented, it is not clear that that the FTIR is the clear better answer for low IWVs. It is clear that the GPS disagrees with the MW and FTIR data, but it could also be concluded that the MW data performs better at low IWV since the variability of both the FTIR-MW and FTIR-GPS data show similar values. This could imply that the FTIR is the root of cause of the large variability. The authors should provide clearer justification for why they conclude FTIR is the method of choice for low IWV."

AC: We made a conclusion of FTIR as a better choice in dry atmosphere not only from
analysis of the scatters between different techniques but also taking into account that RPG-HATPRO radiometer was operating at its limits below 5 mm of IWV due to intrinsic relative weakness of the 22 GHz water vapour line. The uncertainty of MW measurements at these IWV values may reach 10-20%, whereas for FTIR measurements the retrieval errors total 2-4%.

RC: "The approach used in Section 3.3 is not providing an accuracy assessment as much as a relative uncertainty between the three methods. The terminology used in the summary as it being an empirically based upperbound of statistical measurement errors is much more correct."

AC: We corrected the terminology.

RC: "For the uncertainty assessment, selection of dates for which the IWV varied by small amounts (

Are the estimated uncertainties acceptable, do they agree with past work, are there recommendations for which method is more suitable? These questions are covered with the results in the paper, but are not easily found. One last comment is that the authors have an opportunity to discuss whether all three methods can be viewed as equally suitable for IWV and do not. That is, do the authors recommend that a network of IWV instrumentation rely on a single measurement approach or can it use any of the three methods? Is the capability of MW and GPS measuring under cloudy conditions worth the added scatter in the measurments? Is a zenith-viewing MW measuremnt with higher scatter better than a solar-path based FTIR measurement with lower scatter? Including such insight into the paper would improve its relevance to the broader community and take it past being only another collection of IWV retrieval results."

AC: The different observation techniques complement each other rather than outperform each other. We cannot recommend any instrument or technique as the reference one for the networks measuring IWV under variety of atmospheric conditions. FTIR method is highly accurate. MW and GPS are all-weather methods. Based on our results, we propose to use FTIR as a reference method under clear-sky conditions since it is reliable on all scales of IWV variability. The Summary has been revised to address these questions.

We thank the reviewer once more for his independent opinion and suggestions to the text of the manuscript.